# Differential Expression of miRNAs and Behavioral Change in the Cuprizone-Induced Demyelination Mouse Model

**DOI:** 10.3390/ijms21020646

**Published:** 2020-01-18

**Authors:** Seung Ro Han, Yun Hee Kang, Hyungtaek Jeon, Suhyuk Lee, Sang-Jin Park, Dae-Yong Song, Sun Seek Min, Seung-Min Yoo, Myung-Shin Lee, Seung-Hoon Lee

**Affiliations:** 1Eulji Biomedical Science Research Institute, Eulji University School of Medicine, Daejeon 34824, Korea; srhan@eulji.ac.kr (S.R.H.); yhkang@eulji.ac.kr (Y.H.K.); smyoo@eulji.ac.kr (S.-M.Y.); 2Department of Microbiology and Immunology, Eulji University School of Medicine, Daejeon 34824, Korea; jhtjoy0405@gmail.com (H.J.); simpson214@naver.com (S.L.); 3Department of Anatomy and Neuroscience, Eulji University School of Medicine, Daejeon 34824, Korea; bright11star@naver.com (S.-J.P.); dysong@eulji.ac.kr (D.-Y.S.); 4Department of Physiology and Biophysics, Eulji University School of Medicine, Daejeon 34824, Korea; ssmin@eulji.ac.kr; 5Department of Neurosurgery, Eulji University School of Medicine, Daejeon 34824, Korea

**Keywords:** cuprizone, mouse, brain, behavior, miRNAs

## Abstract

The demyelinating diseases of the central nervous system involve myelin abnormalities, oligodendrocyte damage, and consequent glia activation. Neurotoxicant cuprizone (CPZ) was used to establish a mouse model of demyelination. However, the effects of CPZ on microRNA (miRNA) expression and behavior have not been clearly reported. We analyzed the behavior of mice administered a diet containing 0.2% CPZ for 6 weeks, followed by 6 weeks of recovery. Rotarod analysis demonstrated that the treated group had poorer motor coordination than control animals. This effect was reversed after 6 weeks of CPZ withdrawal. Open-field tests showed that CPZ-treated mice exhibited significantly increased anxiety and decreased exploratory behavior. CPZ-induced demyelination was observed to be alleviated after 4 weeks of CPZ treatment, according to luxol fast blue (LFB) staining and myelin basic protein (MBP) expression. miRNA expression profiling showed that the expression of 240 miRNAs was significantly changed in CPZ-fed mice compared with controls. Furthermore, miR-155-5p and miR-20a-5p upregulations enhanced NgR induction through Smad 2 and Smad 4 suppression in demyelination. Taken together, our results demonstrate that CPZ-mediated demyelination induces behavioral deficits with apparent alterations in miRNA expression, suggesting that differences in miRNA expression in vivo may be new potential therapeutic targets for remyelination.

## 1. Introduction

Demyelination of the central nervous system (CNS) is a typical feature of diseases such as multiple sclerosis (MS) and contributes to axon injury and cerebral atrophy, which are characteristic of late stages of the disease [1,2]. MS lesions are pathologically divided into four distinct patterns (I–IV) based on complement activation, IgG deposition, and loss of myelin-associated glycoproteins [3,4]. Spontaneous remyelination comes after demyelination, but often fails to complete [5,6]. Thus, the challenges of MS research are analyzing the causes of remyelination failure and developing methods to restore myelin. The research includes using animal models to help understand the demyelination and remyelination mechanisms, facilitating the study of cellular responses taking place in this process, and providing a robust platform for elucidating putative therapeutic targets. However, there is the limitation that no current animal model faithfully replicates the myriad of symptoms seen in the clinical condition of MS [2,7].

Cuprizone (bis-cyclohexanone-oxaldihydrazone, CPZ) is a copper chelating reagent that can be mixed with a rodent’s normal diet. Continuous feeding of CPZ results in a pathologic pattern similar to that of MS III lesions in white matter. It is, therefore, useful for studying the pathogenesis of primary demyelination due to mitochondrial dysfunction [4,8,9,10]. Completion of demyelination depends on mouse strain [11,12,13], anatomic location [14,15], age [16], dose [17], and sex [12], as in other animal models. However, the pathologic response that follows the CPZ intoxication of C57BL/6 mice is highly reproducible and well-characterized. In this model, demyelination after 4–6 weeks of intoxication is evident in multiple structures, including the hippocampus [18], external capsule [19], rostral cerebellar peduncles [20,21], cerebellum [22,23], striatum [19], cerebral cortex [11,14], and most notably, the corpus callosum [24]. Interestingly, spontaneous remyelination begins at the apex of demyelination and proceeds with the removal of the intoxicant CPZ, resulting in almost complete remyelination within a week [24,25]. As remyelination protects against neurodegeneration and promotes functional recovery [26,27,28], many studies have used the CPZ model to analyze the molecular mechanisms involved in remyelination [29,30,31]. However, the exact mechanisms underlying CPZ-induced demyelination and remyelination, the changes in miRNA expression, and the effects of CPZ treatment on behavior remain to be elucidated. To use the analysis of the miRNA expression in the demyelination model, we rigorously validated several behavioral tests. Here, we observed behavioral changes in mice subjected to a diet containing 0.2% CPZ for 6 weeks, followed by 6 weeks of recovery. Moreover, the expression of 240 miRNAs was significantly changed in CPZ-fed mice compared with control mice. Our results about the change of miRNA expression in vivo might suggest new potential remyelination therapeutic targets.

## 2. Results

### 2.1. Cuprizone Causes Weight Loss Which Subsides after Returning to a Normal Diet

The body weight of the mice was measured weekly throughout the experimental period (Figure 1A). After 1 week of CPZ administration, the treated animals had lost 4.55% of their body weight, whereas those on a normal diet had gained 3.64% (Figure 1B). The difference between the control and the treated mice was statistically significant (*p* < 0.05). After 6 weeks of treatment, despite an overall increase in weight, CPZ-fed mice weighed significantly less than control mice (Figure 1B; *p* < 0.01). Upon returning the CPZ-fed mice to a normal diet, they gained weight rapidly, and after 1 week, were no longer different from the control mice. At the end of the recovery period with normal chow, the treated animals had an overall increase in body weight of 27.27% with respect to the initial weight, while the weight of the control animals increased by 28.5% (treated animals versus controls, *p* = 0.4759, Figure 1B). No deaths occurred during the experimental procedures.

### 2.2. Cuprizone Increases the Number of Falls in the Rotarod Test

Motor coordination was evaluated using a Rotarod apparatus. All animals improved their skills and learned to stay on the rotating rod. At the week before CPZ feeding, there was no difference in the number of falls between the two groups before CPZ feeding (Figure 2A,F). At 6 weeks after CPZ feeding, a two-way repeated-measures analysis of variance showed that there was an interaction between the trial and treatment groups in the number of falls on the first day of training (Figure 2B, *p* < 0.01), indicating different rhythms of learning for the different groups. Animals treated for 6 weeks fell more than control animals on the first trial (Figure 2B,G). Upon returning the CPZ-fed mice to a normal diet, they fell less often than the control mice, and this effect depended on the recovery time (Figure 2C–E,H–J)). These results suggest that 6 weeks of CPZ treatment impaired motor coordination, which was fully recovered after 6 weeks of CPZ withdrawal.

### 2.3. Cuprizone Decreases the Grip Strength of Mice

To evaluate the effect of CPZ on neuromuscular functions, we performed the grip strength test included in the functional observational battery (FOB) commonly used to screen for neurobehavioral toxicity. At the week before CPZ feeding, the grip strength was no different between the control group and CPZ group (Figure 3A,F). The grip strength of mice fed CPZ for 6 weeks was significantly decreased compared to that of control mice (Figure 3B,G). Returning the CPZ-fed mice to normal chow led to recovery of the grip strength to levels similar to those in the control group (Figure 3C–E,H–J). These results suggest that 6 weeks of CPZ treatment induced neuromuscular weakness, which was overcome after 6 weeks of CPZ withdrawal.

### 2.4. Cuprizone Decreases the Total Distance Traveled by Mice in the Open-Field Test

Experimental animals were assessed in the open-field test to evaluate their level of activity and their behavioral response to a novel environment. At the week before CPZ feeding, the total distance traveled was similar between the control group and the CPZ group (Figure 4A). In mice fed CPZ for 6 weeks, the total distance traveled, a measure of general locomotor activity, decreased compared to that in control mice fed normal chow (Figure 4B). The total distance traveled by the treated animals recovered 2 weeks after returning to normal chow (Figure 4C), and increased until 6 weeks after CPZ withdrawal (Figure 4D,E). The habituation pattern was measured for 5 min in the open field (Figure 4F). The habituation patterns of CPZ-fed mice were different from those of the control animals, as shown by the significant interaction between CPZ treatment and the distance traveled. In addition, 6 weeks after returning the CPZ-fed mice to a normal diet, they traveled in the central area of the open field more than the control mice, indicating diminished anxiety (Figure 4F). Furthermore, the heatmap visualizations were similar to the above results (Figure 4G).

### 2.5. Cuprizone Induces a Loss of Myelin in the Corpus Callosum

The corpus callosum (CC) is the largest white matter (WM) structure in the brain, connecting the homologous cortical areas of the two cerebral hemispheres and playing a critical role in the transfer of sensory, cognitive, and motor information [32]. We then focused on the corpus callosum region as one of the most widely studied regions in this model of demyelination. To analyze the loss of myelin in the corpus callosum of CPZ-fed mice, luxol fast blue (LFB) staining, and myelin basic protein (MBP) immunohistochemistry were performed. Brain sections of each group were stained with LFB, which stains the lipid-rich myelin blue (Figure 5A), and with antibodies against MBP, which bind to the protein myelin (Figure 5C). At the beginning of the experiment, the corpus callosums of mice was well-myelinated, with characteristic rows of interfascicular oligodendrocytes (Figure 5A, left panels). Strong loss of myelin was detected in the mice fed the CPZ diet at 4 weeks (Figure 5A, middle panels), which was slightly overcome at 6 weeks of CPZ treatment (Figure 5A, right panels). Figure 5B revealed these integral optical densities (IOD). A similar demyelination trend was observed on MBP immunohistochemical analysis (Figure 5C) and IOD (Figure 5D). These results demonstrate that CPZ alters the myelin lipid and protein structure, resulting in motor neurobehavioral defects.

### 2.6. Diverse Expression Patterns of microRNAs in Cuprizone-Induced Demyelination

Demyelination of the central nervous system (CNS) is the hallmark of diseases such as multiple sclerosis (MS), an immune-mediated inflammatory disease of the central nervous system. miRNAs play important roles in autoimmune diseases, including MS. Although miRNA expression profiles of MS patients were recently reported [33,34], the role of these molecules in MS remains to be elucidated.

To evaluate the miRNA expression pattern in CPZ-induced demyelination, we used the Affymetrix GeneChip^®^ miRNA 4.0 array. We identified 1348 and 1253 expressed miRNAs in control and CPZ-treated mice, respectively. Furthermore, the expression of 240 miRNAs was significantly changed in CPZ-treated mice compared with controls—these were clustered by hierarchical clustering to create the heatmap shown in Figure 6A. The top five miRNAs by fold change vs. the average expression in CPZ-fed mice compared with control mice were mmu-miR 146a-5p, mmu-miR-20a-5p, mmu-miR-338-5p, mmu-miR-145a-5p, and mmu-miR-219a-2-3p (Figure 6B). Figure 6C represents the miRNA expression heatmap, including the top five miRNAs, using the MORPHEUS tutorial program. The expressions of miR 146a-5p, miR 155-5p, and miR 20a-5p were upregulated in CPZ-treated mice compared with control mice. However, the miR 145a-5p, miR 338-5p, and miR 219a-2-3p were downregulated in CPZ-treated mice (Figure 6C). We selected the 39 (30) miRNAs that were upregulated (downregulated) in CPZ-fed mice with a fold change (FC) of ≥1.5 compared with control mice and predicted their respective target genes using the micro-RNA database (miRDB) (Table 1 and Table 2, respectively). This analysis demonstrated the diverse expression patterns of miRNAs in CPZ-induced demyelination. It could be used to clarify the relationship between miRNA modulation and the molecular cascade induced during CPZ-induced demyelination.

### 2.7. Validation of miRNAs Expression Using the miRNA Quantitation PCR

To validate the changes of miRNA expression in cuprizone-induced demyelination mice, we performed the miRNA quantification polymerase chain reaction (qPCR), including the top five miRNAs by fold change vs. the average expression in CPZ-fed mice compared with control mice and miR 155-5p (Figure 7). As shown in Figure 7A–C, the expressions of miR 146a-5p, miR 155-5p, and miR 20a-5p were upregulated in demyelinated mice (D6W) as in the results by miRNA microarray, and then downregulated at 6 weeks after cuprizone withdrawal (R6W) compared to control mice (Cont). The miR 145a-5p and miR 219a-2-3p were downregulated in D6W, and confirmed miRNA microarray (Figure 7D,E), but not miR 338-5p (Figure 7F); however, three miRNAs were downregulated R6W compared to Cont. Then, to focus on the regulation mechanism of de- and remyelination, we analyzed the expressions of the miRNAs’ predicted target gene (Table 1 and Table 2) using real-time PCR (Appendix A) and western blotting (Figure 8).

### 2.8. Analysis of miRNA Target Genes and Signal Cascades in Cuprizone-Induced Demyelination

In our previous study, we showed the activation of Smad 2, 3, and 4 via the TGFβ pathway in glioblastoma multiforme and primary glioma stem cells and discussed the importance of the smad pathway in brain diseases [35]. We hypothesized that the smad pathway should be one of the regulation pathways of demyelination. Therefore, we focused on the upregulated miR 155-5p and miR 20a-5p, which predicted to target Smad 2 and Smad 4/TGFβR2 in cuprizone-fed mice (Table 1).

To evaluate the TGFβ pathway such as TGFβR1, which was a predicted target of miR 20a-5p, TGFβR2, and TGFβ1 and its signal cascade genes, Id1 and Nogo receptor (NgR) in CPZ-induced demyelination and remyelination after CPZ withdrawal, we carried out the real-time PCR (Appendix A). The transcriptional levels of TGFβ1, TGFβR1, and TGFβR2 were not significantly altered in CPZ-induced demyelination (Appendix A), but NgR was significantly increased (Appendix A). These results showed that the upregulated miR 20a-5p could not affect upstream molecules of smad signal cascades, but finally regulated the downstream molecules of smad signal cascades.

Therefore, we evaluated the smad family and Id1 protein, the predicted targets of upregulated miR 155-5p and miR 20a-5p using western blot analysis (Figure 8). As shown in Figure 8, the expression and phosphorylation of Smad 2 and 3 were dramatically decreased in CPZ-induced demyelination mice compared to control mice (Figure 8A,B). Also, Smad 4 and Id1 expression were downregulated in demyelination (Figure 8A,B). Thus, these results demonstrate that miR 155-5p and miR 20a-5p upregulation in demyelination leads to the induction of NgR expression through the suppression of Smad signal cascades. We then proposed this as one of the modulated mechanisms of demyelination (Figure 9).

## 3. Discussion

The disruption of the myelin sheath of CNS is a prominent phenomenon of many clinically relevant disorders on the basis of physiopathology [36]. Several mouse models are available for studying demyelination and remyelination. The best-characterized demyelinating mouse model is the C57BL/6J with 0.2% CPZ added to the diet [24]. Several hypotheses have been proposed [17,24,37], but it is still unclear why CPZ, a copper chelator molecule, specifically affects oligodendrocytes, the cell type that synthesizes and maintains the CNS myelin sheath.

The behavior of CPZ-fed mice has not been sufficiently studied [17,37,38]. Morell et al. (1998) reported that CPZ-treated animals did not appear to be as active at the end of the first week of CPZ administration, and also during the following several weeks of CPZ exposure. The same authors reported that, during the recovery period, the animals displayed normal activity levels, and were indistinguishable from control animals [37]. However, Franco-Pons et al. (2007) reported that behavioral deficits followed the course of demyelination–remyelination induced by administration of 0.2% CPZ and that some of the changes persisted even after 6 weeks of a normal diet [38]. Our experiments showed motor neurobehavioral defects using the Rotarod, grip strength, and open-field tests after 6 weeks of 0.2% cuprizone feeding, and the recovery of such neurobehavioral defects upon returning to a normal diet for 6 weeks, together with remyelination. We believe the differences in the behavioral patterns observed in the various studies might have been caused by differences in the behavioral assessment protocols, underlining the importance of using standardized behavioral assessment protocols.

miRNAs are short, noncoding RNA molecules that are processed from larger transcripts of non-classical genes by the Drosha and Dicer nucleases [39]. Mature miRNA biogenesis is a tightly controlled, multistep process finalized in the production of a ~22-nt-long duplex [40]. miRNAs regulate gene-expression programs by reducing the translation and stability of target mRNAs [41]. It has been estimated that the expression of as many as one-third of protein-coding genes may be regulated by miRNA [42].

Many miRNAs have functioned as important regulators of the immune system [33,34,43,44,45,46]. In particular, miR-155 is known to be a crucial regulator orchestrating the role of numerous acquired and native immune cell populations. The miR-155 host gene (MIR155HG) produces miR-155-3p and miR-155-5p, being the functional form [47]. Several researchers identified increased miR-155 expression in T cells both in vivo and in vitro during the development of autoimmune responses [48], and miR-155-deficient mice have been found to be resistant to the development of experimental autoimmune encephalomyelitis (EAE) [49]. Additionally, Mycko et al. (2015) identified an EAE-specific mechanism of miR-155 expression, in which miR-155-3p drives the development of autoimmune demyelination by regulating heat shock protein 40 [46]. Furthermore, we showed that miR-155-5p, the functional form of miR-155, is significantly upregulated in mice subjected to CPZ-induced demyelination. This result is in line with those of other reports described above, and suggest that miR-155-5p upregulation may induce demyelination through upregulation of NgR via suppression of Smad signal cascades in CPZ-fed mice. However, in Table 1, there is mmu-miR-195a-3, which might suppress Id1 and increase with demyelination. Proteins Smad2, Smad3, and Smad4, as seen from the tables, are also inhibited by various miRNAs. We would not ignore such direct connections between the Smad family and various miRNAs.

One of the MS-related miRNAs, miR-146a, is differentially expressed in MS lesions and promotes the differentiation of oligodendrocyte precursor cells (OPCs) during remyelination [50]; moreover, it is upregulated during CPZ-induced de- and remyelination [51]. In particular, Martin et al. (2018) reported that experimental demyelination and axonal loss are reduced in mice deficient in microRNA-146a and that the number of OPCs is slightly higher in WT mice during remyelination, indicating the complex role of miR-146a during in vivo de- and remyelination [51]. In our results, miR-146a-5p was significantly upregulated in mice subjected to CPZ-induced demyelination, suggesting that it may be a key miRNA involved in the induction of demyelination in CPZ-fed mice. These results support the reliability of our miRNA array results, which may be useful in understanding the demyelination mechanism. However, a limitation of our study is that the miRNA expression pattern was not analyzed in remyelinated mice. Thus, we cannot thoroughly discuss the changes in miRNA expression during CPZ-induced de- and remyelination.

It is a long-held view that the motor corpus callosum is important for bimanual coordination and learning of bimanual motor skills. This notion was built mainly on bimanual coordination deficits observed in patients with extensive lesions of the corpus callosum or partial callosotomies [52]. The administration of CPZ in combination with 0.5 mg/kg of WIN55212.2 declined the expression of NgR1 and might confer neuroprotection against CPZ [53]. Actually, NgR1 participates in oligodendrocyte differentiation, myelination [54], and cell cytoskeleton reorganization [55]. Our supplementary data shows that the expression of NgR increased at 6 weeks after CPZ treatment in the corpus callosum, suggesting that the neuroprotection in axon fibers against CPZ and the participation of OPC differentiation into myelinating oligodendrocytes resulted in spontaneous remyelination in CPZ-fed mice. These results support the reliability of our predicted gene targets in miRNA microarray results and provide a better understanding of the demyelination mechanism. Based on the findings, we are looking forward to developing the new targets underlying the de- and remyelination mechanism as a useful therapeutic target for remyelination.

## 4. Materials and Methods

### 4.1. Animals and Cuprizone Administration

Twenty-six male C57BL/6 mice aged 7 weeks were purchased from KOATECH (Pyeongtaek, Korea) and allowed 1 week of acclimation to their new environment before beginning the experiment. The 0.2% CPZ diet (TD.140803) and normal diet (2018S) were purchased from ENVIGO (Madison, WI). The mice in the CPZ group were freshly fed the 0.2% CPZ diet each day for a total of 6 weeks to induce demyelination. The control mice, kept on a normal diet, were age-matched to the CPZ-fed mice. Food pellets were changed every day. After 6 weeks of CPZ treatment, the diet was changed to normal rodent chow for an additional 2, 4, and 6 weeks to examine remyelination. According to a bibliographic investigation before the start of the experiment [13,38,51,56], it was confirmed that both females and males were used to induce the cuprizone-induced model, and the test results were similar in the laboratory. Twenty-six animals were divided into two groups of 13 animals each. The final data was made with the results of the total individuals minus the number of sacrificed mice. N = 7 in each group. These were sufficient numbers for different aspects of the experiment. All animal protocols adhered to the Ministry of Food and Drug Safety (MFDS) Guidelines for Care and Use of Laboratory Animals and were approved by the Eulji University Laboratory Animal Care and Use Committee (Approval No., EUIACUC17-21; approval date, 15 December 2017).

### 4.2. Motor Coordination and Learning: Rotarod Test

Motor coordination and balance were evaluated in a Rotarod apparatus (BR1001, B.S Technolab INC., Seoul, Korea), which consists of a motor-driven rotating rod whose speed can be adjusted. All mice were evaluated on the Rotarod three times a day for two consecutive days, with the rotation set at 15 to 16 revolutions per minute (rpm). To test the performance, the mice were placed on the rotating cylinder at an angle of 45° with an initial rotation speed of 16 rpm, and were allowed to run for 60 s. If the animal fell, the chronometer was stopped, the animal was put back onto the cylinder, and then the timing was continued. The trial was repeated after 5–10 min. The falls and flips (when the animal hangs on to the cylinder and continues all the way around) were recorded within 60 s of each trial.

### 4.3. Grip Strength Test

The grip strength test allows for the assessment of neuromuscular functions by determining the maximal peak force developed by a rodent when the operator tries to pull it out of a specially designed grid, available for both the fore and hind limbs. The grip strength test is included in the functional observational battery (FOB) used to screen for neurobehavioral toxicity. In this context, changes in grip strength peak values of the rodents are interpreted as evidence of motor neurotoxicity. Forelimb strength (g force) was measured with a grip strength tester (BIO-GS3, BIOSEB, Vitrolles Cedex, France) to detect contralateral paw weakness. After both forelimbs of the mouse were loosened by pulling the tail, the maximal force was recorded. Each mouse was subjected to three trials of each test, and the mean values (g) were calculated and normalized to body weight.

### 4.4. Open-Field Assessment

Activity and behavioral responses to a novel environment were measured in an open-field apparatus consisting of a 30 cm × 30 cm wooden square surrounded by a dark wall 30 cm high. The mice were placed in the center of the arena at the beginning of the test period. During the test, the mice were allowed to move freely around the open field and explore the environment for 5 min. Their path was recorded by a video camera (LS903, LG Electronics, Daejeon, Korea) placed above the square. The video tracking program EthoVision XT5 (Noldus Information Technology, Wageningen, NL, Canada) was used to measure the total distance traveled.

### 4.5. Luxol Fast Blue Staining and Myelin Basic Protein Immunohistochemistry

All mice were deeply anesthetized by intraperitoneal injection of ketamine (80 mg/kg) and xylacine (10 mg/kg) dissolved in 0.9% saline and transcardially perfused with 4% paraformaldehyde (PFA). For Luxol fast blue staining and immunohistochemical analysis, the brains were placed in 30% sucrose/phosphate buffered solution (PBS) for 48 h at 4 °C and then snap-frozen in isopentane. The corpus callosum was chosen as a representative white matter region as it has been extensively examined in this animal model [17,37]. Coronal sections (30 μm thick) of the mouse brain, corresponding to bregma 0.62 mm to bregma -0.46 mm, were cut with a cryostat and stored at −20 °C in cryoprotection buffer (40% phosphate buffer 0.1 M, 30% glycerol, 30% ethylene glycol).

Demyelination was determined histochemically by examining coronal sections stained with Luxol fast blue (LFB). For each staining, three sections per animal were used, and a total of two mice were tested for each group. The staining was done at least twice each. All reagents were purchased from Abcam (ab150675, Cambridge, UK). Luxol fast blue staining was carried out in accordance with the manufacturer’s instructions (Abcam, Cambridge, UK). Brain tissue slides were incubated in Luxol Fast Blue Solution (0.1%) for 24 h at room temperature. The sections were rinsed thoroughly in distilled water, and differentiated by dipping in Lithium Carbonate Solution (0.05%) several times. Differentiation for the sections were then continued by repeatedly dipping in alcohol reagent (70%) until the gray matter became colorless and the white matter remained blue. Then, the sections were rinsed in distilled water, followed by incubation with Cresyl Echt Violet (0.1%) for 2–5 min. The sections were rinsed quickly in one change of distilled water, dehydrated quickly in three changes of absolute alcohol, cleared in three changes of xylene, and mounted with mounting medium.

The method used for myelin basic protein immunohistochemical analysis has been described in a previous article [57]. For detection of MBP expression, we used immunodetection kits (M.O.M. kit, MP-2400, Vector laboratories, Burlingame, CA, USA) which are specifically designed to localize primary mouse antibodies on mouse tissues. Subsequently, all sections were treated with 3% H_2_O_2_ for 10 min to block endogenous peroxidase activity, and then slides were incubated for 1 h in mouse Ig blocking reagent (Vector laboratories, Burlingame, CA, USA). Then, the slides were blocked with 2.5% normal horse serum (Vector laboratories, Burlingame, CA, USA) for 1 h at room temperature (RT) and then incubated with anti-MBP antibody (1:100; sc-271524, Santa Cruz Biotechnology, Inc., Santa Cruz, CA, USA) overnight at 4 °C. The sections were then incubated with the Immpress anti-mouse IgG reagent (made in horse; Vector laboratories, Burlingame, CA, USA) for 30 min at RT. The second Ab used in this study, horse anti-mouse IgG, had novel conjugation and micropolymer chemistries to create a highly sensitive, ready-to-use, one-step, non-biotin detection system for immunohistochemistry staining. The color was then developed with 3,3′-diaminobenzidine (DAB) and counterstained with Mayer’s hematoxylin. The DAB developing time should be determined by the investigator under a microscope, but generally, optimal staining intensity is produced in 2–10 min. DAB incubation on the slides was about 2 min in this staining. Finally, the slides were observed under an Eclipse E400 microscope (Nikon Instruments Inc., Melville, New York, NY, USA) and images were captured with a Nikon Digital Sight DS-U2 camera. The integral optical density (IOD) of LFB and MBP were analyzed by *i*-SOLUTION software (Image & Microscope Technology Inc., Burnaby, BC, Canada) by extracting nuclei from photos [58]. For Luxol fast blue stained sections, the area of the deep sky-blue signal and its intensity were assessed using the IOD analyzed by *i*-SOLUTION software. The brown coloring was regarded as a positive signal of MBP immunoreactivity, and its intensity of MBP immunoreactivity was evaluated by IOD. Nuclei were excluded from the analysis of IOD. Each section taken to obtain a mean optical density value had images at 400× magnification.

### 4.6. RNA Isolation and RNA Quality Check for Affymetrix miRNA Arrays

Total RNA was extracted from the cerebrum of normal/CPZ-paired mice using the easy-BLUE^TM^ Total RNA Extraction Kit (iNtRON biotechnology, Daejeon, Korea) according to the manufacturer’s instructions. RNA purity and integrity were evaluated using an ND-1000 Spectrophotometer (NanoDrop, Wilmington, NC, USA) and Agilent 2100 Bioanalyzer (Agilent Technologies, Palo Alto, CA, USA), respectively.

### 4.7. Affymetrix miRNA Arrays

Affymetrix Genechip miRNA 4.0 array processing was performed according to the manufacturer’s protocol. RNA samples (1000 ng) were labeled with the FlashTag™ Biotin RNA Labeling Kit (Genisphere, Hatfield, PA, USA). The labeled RNA was quantified, fractionated, and hybridized in the miRNA microarray according to the standard procedures recommended by the manufacturer. The labeled RNA was heated to 99 °C for 5 min and then to 45 °C for 5 min. RNA-array hybridization was performed with agitation at 60 rotations per minute for 16 h at 48 °C on an Affymetrix^®^ 450 Fluidics Station. The chips were washed and stained using a Genechip Fluidics Station 450 (Affymetrix, Santa Clara, CA, USA). The chips were then scanned with an Affymetrix GCS 3000 scanner (Affymetrix). Signal values were computed using the Affymetrix^®^ GeneChip™ Command Console software.

### 4.8. Raw Data Preparation and Statistical Analysis of Affymetrix miRNA Arrays

Raw data were extracted automatically with the Affymetrix data extraction protocol using the Affymetrix GeneChip^®^ Command Console^®^ Software (AGCC) software. The CEL files and miRNA level quantified by RMA+DABG-All analysis were imported, and the results were exported using Affymetrix^®^ Power Tools (APT) software. The array data were filtered to select probes annotated to the mouse genome. The comparative analysis between the test sample and the control sample was based on fold-change. Hierarchical cluster analysis was performed on the differentially expressed miRNAs using complete linkage and the Euclidean distance as a measure of similarity. Statistical tests and the visualization of differentially expressed genes were conducted using the R statistical language v. 3.1.2 (https://www.r-project.org/). In addition, miRNA expression was analyzed using the MORPHEUS tutorial program (MACROGEN, Seoul, Korea).

### 4.9. Statistical Analysis

Data are shown as mean ± standard deviation, and significant differences were assessed using the one-way analysis of variance (ANOVA) with the post hoc test and student’s *t*-test using SPSS for Windows, release 17.0 (SPSS Inc., Chicago, IL, USA). Repeated-measure analysis of variance was also used to evaluate accustomization in the open-field test, the rota-rod test, and grip strength test. The level of statistical significance stated in the text is based on the *p*-values * *p* < 0.01, ** *p* < 0.005, or *** *p* < 0.001 which were considered statistically significant.

### 4.10. Real-Time PCR for miRNA Quantitation and Target Genes of miRNAs

Oligonucleotide sequences corresponding to the miR146a-5p, miR20a-5p, miR155-5p, miR145a-5p, miR219a-2-3p, and miR338-5p were mimicked on miRDB sequences, and the TGFβ1, TGFβR1, TGFβR2, Id1, and NgR gene were designed using Primer3 software (http://frodo.wi.mit.edu). The first-strand cDNA mixture contained 1.0 μg of total RNA as a template for miRNA quantitative PCR. The Mir-X^TM^ miRNA First-strand Synthesis and TB Green^TM^ kits (TaKaRa, Daejeon, Korea) were used to perform real-time PCR according to the manufacturer’s protocol. All primer sequences have been described in Appendix A. Optimized real-time PCR was carried out as a previous study [35]. Relative miRNA expression levels were nomalized to U6 expression, and relative gene expression levels were normalized to GAPDH expression.

### 4.11. Western Blot Analysis

Western blot analysis of miRNA target proteins was performed according to the previous report [35]. Sodium dodecyl sulfate-polyacrylamide gel electrophoresis (SDS-PAGE) was conducted using a Mini-PROTEIN^®^ System (Bio-Rad, Hercules, CA., USA) and a 12% gel according to the manufacturer’s protocol. Proteins were transferred to a nitrocellulose blotting membrane and probed overnight at 4 °C with primary antibodies (1:1000, pSmad 2, Smad 2, pSmad 3, Smad 3, and Smad 4, Bioss; Id1, Santa Cruz; β-actin, Sigma-Aldrich) followed by 1 h at RT by an HRP-conjugated secondary antibody (1:2000, rabbit anti-mouse HRP and goat anti-rabbit HRP, Santa Cruz). Immunolabeled proteins were detected by incubation with enhanced chemiluminescence (ECL) substrate, followed by exposure of the membrane to autoradiography film. 

## 5. Conclusions

In conclusion, our results indicate that behavioral deficits follow the course of demyelination induced by administration of 0.2% CPZ, and that some of the changes do not persist after 6 weeks on a normal diet. In particular, the expression of 240 miRNAs was significantly changed in CPZ-fed mice compared with control mice. In particular, miR 155-5p and miR 20a-5p upregulations may induce demyelination through upregulation of NgR via suppression of Smad signal cascades in CPZ-fed mice. Taken together, these results suggest that the changes in miRNA expression in vivo are new potential remyelination therapeutic targets.

## Figures and Tables

**Figure 1 ijms-21-00646-f001:**
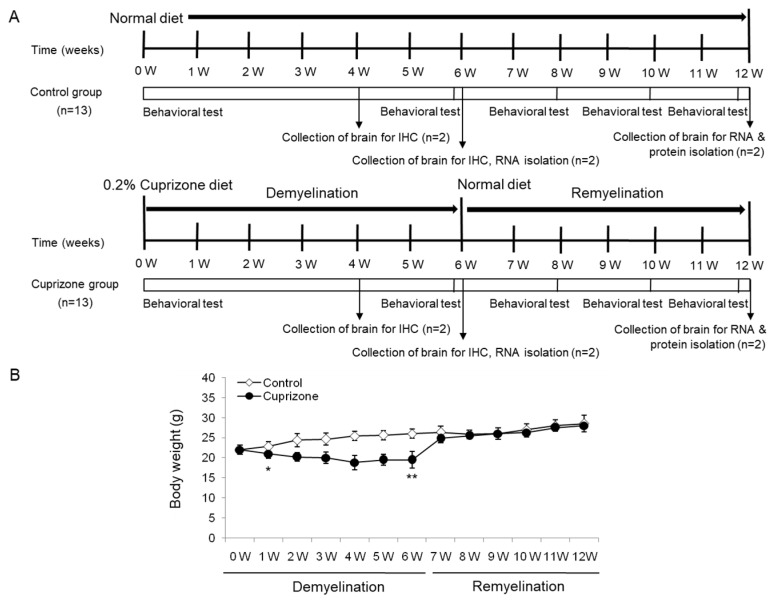
Schematic representation of the experimental protocols and the measurement of body weight. (**A**) The control group was fed normal chow, and two of the thirteen mice were sacrificed after 4, 6, and 12 weeks. The cuprizone group was fed a 0.2% cuprizone (CPZ) diet, and two of the thirteen mice were sacrificed after 4 and 6 weeks, respectively. The CPZ group was fed the CPZ-containing diet for 6 weeks and then allowed to recover for 6 weeks on normal chow without CPZ. Behavior assessments were conducted in the two groups at 0, 6, 8, 10, and 12 weeks during the experimental period, and two of the nine mice were sacrificed at 6 weeks after CPZ withdrawal, respectively. (**B**) The body weight of the mice was measured weekly throughout the experimental period. The final data was made with the results of the total individuals minus the number of sacrificed mice. *N* = 7 for each group, and * *p* < 0.05, ** *p* < 0.01.

**Figure 2 ijms-21-00646-f002:**
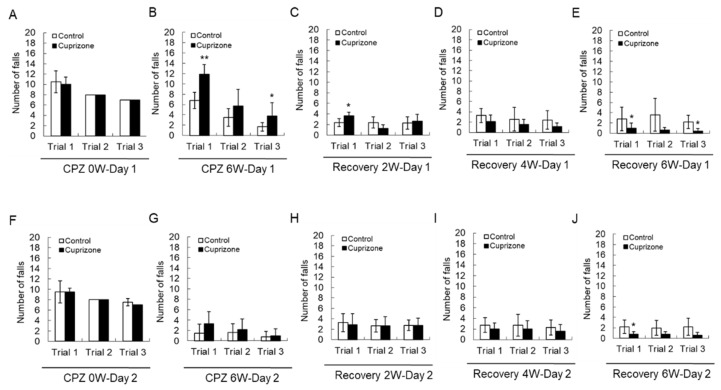
Number of falls in the Rotarod test. (**A**–**E**) The graphs show the number of falls in three different trials on the first day. (**F**–**J**) The graphs show the number of falls in three different trials on the second day. Data are expressed as mean ± standard deviation (SD). The final data was made with the results of the total individuals minus the number of sacrificed mice. *N* = 7 for each group, and differences between groups are expressed as * *p* < 0.05, ** *p* < 0.01.

**Figure 3 ijms-21-00646-f003:**
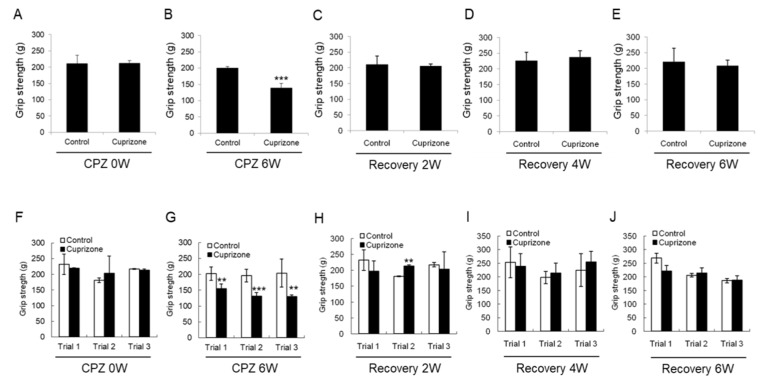
Results of the grip strength test. (**A**–**E**) The graphs show the average grip strength of three different trials in the control or cuprizone group. Data are expressed as mean ± standard error of the mean (SEM). (**F**–**J**) The graphs show the grip strength in three different trials. Data are expressed as mean ± standard deviation (SD). The final data was made with the results of the total individuals minus the number of sacrificed mice. *N* = 7 for each group, and differences between groups are expressed as ** *p* < 0.01, *** *p* < 0.001.

**Figure 4 ijms-21-00646-f004:**
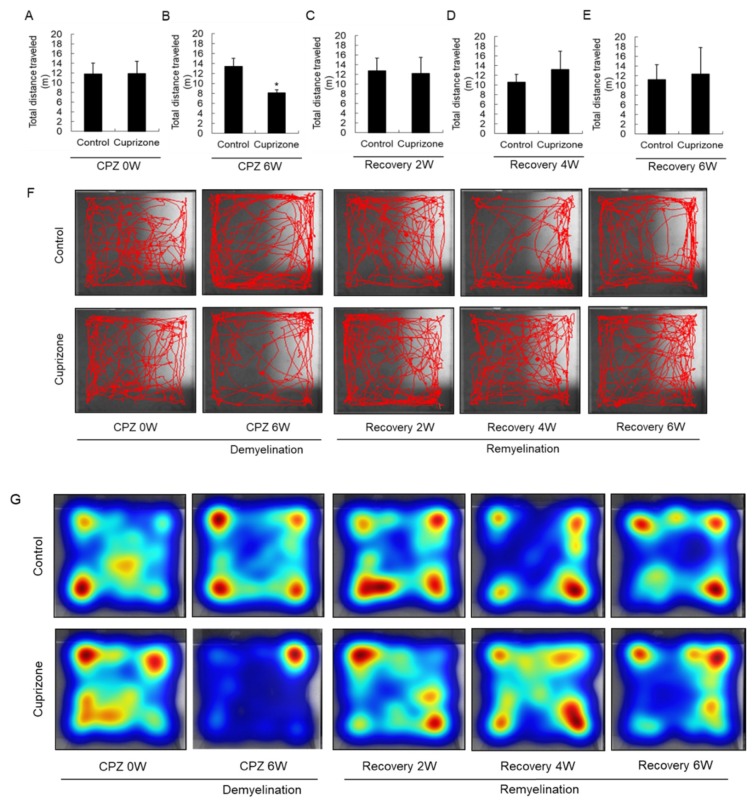
Open-field behavior. (**A**–**E**) The graphs show the total distance traveled in an open field during 5 min. Data are expressed as mean ± standard deviation (SD). The final data was made with the results of the total individuals minus the number of sacrificed mice. *N* = 7 for each group, and differences between groups are expressed as * *p* < 0.05. (**F**) Representative images showing typical examples of exploratory behavior in the open-field test in the control and cuprizone groups. (**G**) The heatmap visualizations were similar to the above results.

**Figure 5 ijms-21-00646-f005:**
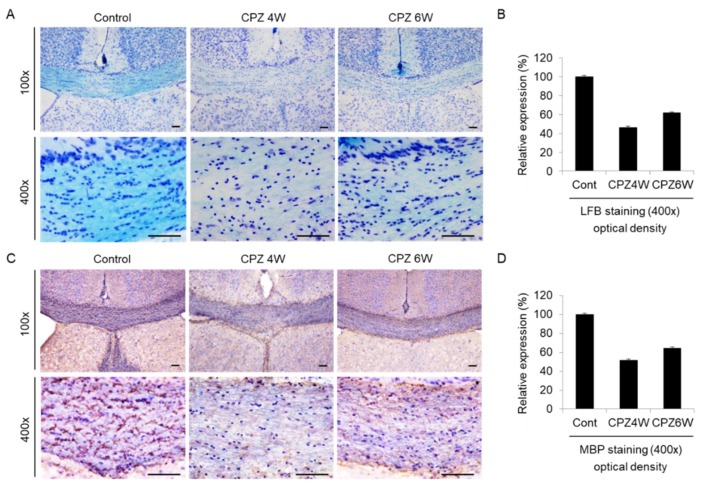
Induction of demyelination in cuprizone-fed mice. (**A**) Luxol fast blue (LFB)/Cresyl Echt Violet staining in coronal slices of the cuprizone-fed and control groups in the corpus callosum. Myelinated fibers are shown in blue, and Nissil substance and nerve cells in violet. Lithium carbonate indicates differentiation, and Cresyl Echt Violet is a counterstain. Scale bar: 50 μm. (**C**) Myelin basic protein (MBP) immunostaining in coronal slices of the cuprizone-fed and control groups in the corpus callosum. Control mice display a normally myelinated corpus callosum. Maximal demyelination is observed after 4 weeks on a cuprizone diet, and the recovery of demyelination is weakly observed after 6 weeks on a cuprizone diet. Scale bar: 50 μm. For each staining, three sections per animal were used and a total of two mice were tested each group. The staining was done at least twice each. (**B**,**D**) The integral optical density of LFB or MBP revealed using *i*-Solution software.

**Figure 6 ijms-21-00646-f006:**
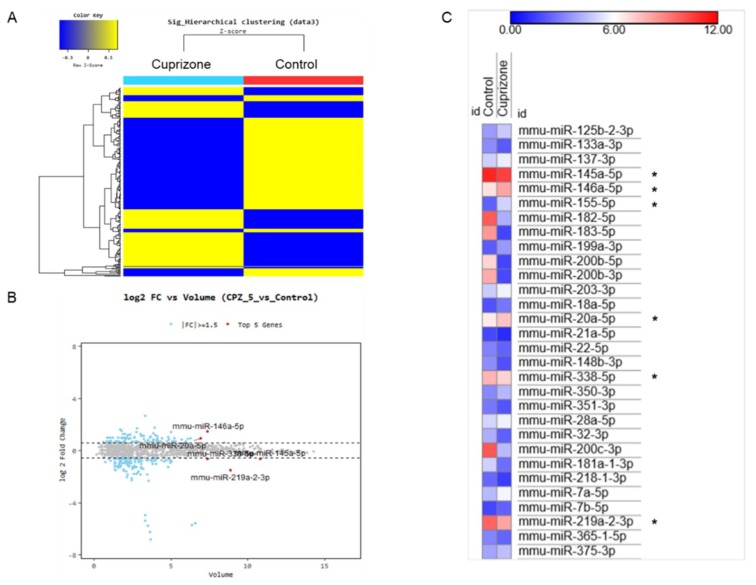
The miRNA expression analysis of cuprizone-fed mice and control mice. (**A**) Hierarchical clustering map of miRNA expression levels in cuprizone-fed mice versus control mice. Color key is Z-score (−0.5~0.5). (**B**) The graph shows the top five miRNAs evaluated using logarithmic fold change (log 2) vs. average expression in cuprizone-fed mice compared with control mice. (**C**) The expression heatmap shows miRNA expression changes (color key is expression score, 0–12), including the top five miRNAs (*) using the MORPHEUS tutorial program.

**Figure 7 ijms-21-00646-f007:**
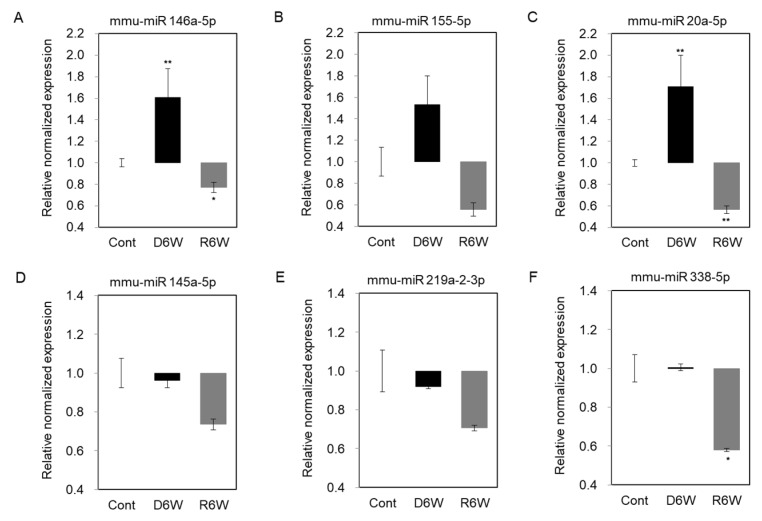
Validation of miRNA expression using miRNA quantitation Polymerization Chain Reation (PCR) in cuprizone-induced demyelination and remyelination mice. (**A**–**C**) The expression levels of miR 146a-5p, miR 155-5p, and miR 20a-5p were significantly upregulated in demyelinated mice (D6W) compared to control mice (Cont) by real-time PCR analysis. (**D**,**E**) The expression levels of miR 145a-5p and miR 219a-2-3p were weakly downregulated in D6W compared to Cont., but not miR 338-5p (F). *n* = 2, in triplicate for each group. The differences between groups are expressed as * *p* < 0.05, ** *p* < 0.01.

**Figure 8 ijms-21-00646-f008:**
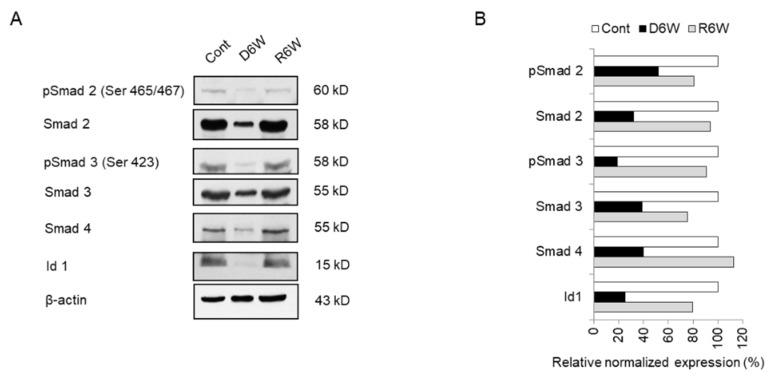
Analysis of miRNA target genes expression in cuprizone-fed mice and control mice. (**A**) Western blot analysis shows the phosphorylated and total Smad 2, Smad 3, Smad 4, and Id1 in demyelinated mice (D6W) and remyelinated mice (R6W) compared to control mice (Cont). *n* = 2. (**B**) The graph shows the relative normalized expression of these in D6W and R6W with Cont. The uncut original western blot images are in Appendix A.

**Figure 9 ijms-21-00646-f009:**
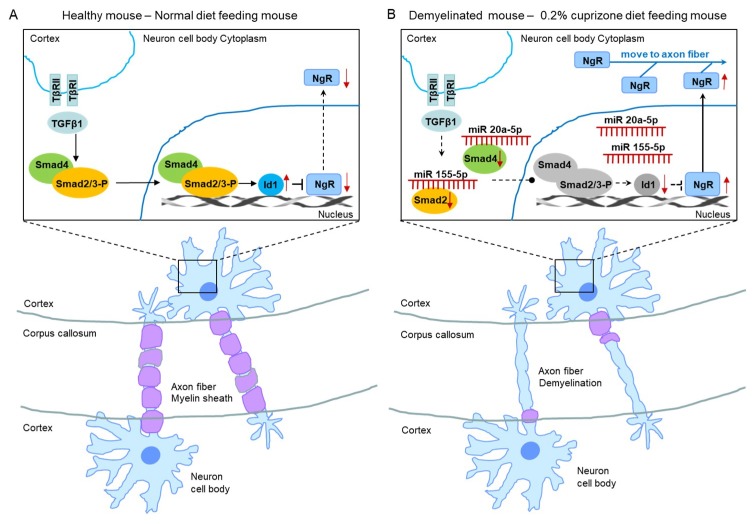
The proposed mechanism of miR 155-5p and miR 20a-5p upregulations led to demyelination. (**A**) The illustration shows a proposed mechanism in normal diet-fed, healthy mice. (**B**) The illustration shows that the proposed mechanism of miR 155-5p and miR 20a-5p upregulations led to demyelination in 0.2% cuprizone diet-fed mice.

**Table 1 ijms-21-00646-t001:** Upregulated miRNA in a cuprizone-induced demyelination mouse model (Fold Change ≥ 1.5).

Transcript ID (Array Design)	CPZ_5/Control.fc	CPZ_5/Control.volume	Predicted Targets in miRDB	Sequence Length	Sequence
mmu-miR-146a-5p	2.707892	7.386574	Nova1, Golph3l, Eif4g2, Slc10a3, Kras, Usp3, Irak1	22	UGAGAACUGAAUUCCAUGGGUU
mmu-miR-20a-5p	1.894162	6.941493	Tnfrsf21, Map3k2, Rab5b, Arhgap12, E2f1, Arhgap1, Tgfbr2, Fzd3/5, Smad4/5/7, CXCL12, Ccnd1, Cadm2	23	UAAAGUGCUUAUAGUGCAGGUAG
mmu-miR-34c-5p	1.919406	6.30073	Vamp2, Notch1, E2f5, Il6ra, Ubl4, Rtn4rl1, Cx3cl1	23	AGGCAGUGUAGUUAGCUGAUUGC
mmu-miR-431-5p	1.836794	6.098394	Zfp644, Smurf1, CD34, Camta1, Cltc, Ctsb, Zeb1, Lamp2, Smad4	21	UGUCUUGCAGGCCGUCAUGCA
mmu-miR-146b-5p	1.586426	5.961954	Nova1, Golph3l, Eif4g2, Bcorl1, Kras, Usp3, Irak1, Psmd3	22	UGAGAACUGAAUUCCAUAGGCU
mmu-miR-203-3p	1.898645	5.216835	Ptp4a1, Gabarapl1, Tbk1, Nova1, Ubr1, Ctss, Rab10, Cdh10, Rtn4(ts88%)	22	GUGAAAUGUUUAGGACCACUAG
mmu-miR-665-3p	1.588351	5.094984	Hars2, Stk35, Il1r1, Abr, Eno2, Ncam1, Syn1, Fgfr1, Edem1, Sumo1	20	ACCAGGAGGCUGAGGUCCCU
mmu-miR-7a-5p	2.651576	5.018355	Sp1, Cdon, Casp9, Ide, Mapkap1, Mobp(94%), Edem1, Vdac1, Mapk8	23	UGGAAGACUAGUGAUUUUGUUGU
mmu-miR-17-3p	1.785589	4.874162	Adam7, Vezf1, Rab35, Neurog1, Apaf1, Hdac3, Golim4, Cdk17, Vim(74%)	22	ACUGCAGUGAGGGCACUUGUAG
mmu-miR-1946b	2.028228	4.81781	Rac2, Ctsa, Cnnm3, IL34, Tnfrsf10b	26	GCCGGGCAGUGGUGGCACAUGCUUUU
mmu-miR-6980-5p	1.623165	4.770035	Pip5k1c, Pgap3,Crispld2, Mcl1, Cadm4, Nova2, Mapkap1, Cdk4	22	GUGGGGGGGGAGGCUAGGUUAG
mmu-miR-7648-3p	1.884263	4.431739	sp1, Cdon, Casp9, Ide, Mapkap1, Mobp(94%), Edem1, Vdac1, Mapk8	21	AGGGCUGGGCCCGGGACGCGG
mmu-miR-6970-5p	1.606978	4.332724	Cldn25, Cstc, Hars2, Tgfb2, Dek, Cd2ap, Ccnc(cyclin c), Paxip1, Pi15, Necap1, Usp3	25	GUAAGUUCAGGGCUGGGAGCAGAGA
mmu-mir-665	1.584525	4.221859	3p:Hars2, Cipc, Tgfb3, Klk9, Cxcl1, Sdc1, Cadm3, Smad7, Tgfbr1, Dnm1	94	AGAACAGGGUCUCCUUGAGGGGCCUCUGCCUCUAUCCAGGAUUAUGUUUUUAUGACCAGGAGGCUGAGGUCCCUUACAGGCGGCCUCUUACUCU
mmu-miR-380-5p	1.54359	4.158481	Frk, Socs2, Snai2, Neurod1, Il33, Nkiras1, Tgfbr2, Sp3, Rb1, Ccr7, smad5, Bmp4	22	AUGGUUGACCAUAGAACAUGCG
mmu-miR-125b-2-3p	2.096336	4.139472	Cd28, Dab2, Stim2, Scai, Frk, Api5, Ret, Nedd9, H3f3b, Gga3, Tab2, Vcam1	22	ACAAGUCAGGUUCUUGGGACCU
mmu-miR-7066-3p	1.705388	4.096965	Tnfaip2, Mtss1, Nrcam, H3f3b, Ergic1, Moap1, Mcam	23	UCUACCCAUUGCCUGCCUCCCAG
mmu-miR-195a-3p	1.690828	4.045272	Mbp(54%), Cnr1, Ube2i, Hif1a, Nedd9, Negr1, Hand2, Cadm2, Id1, Rab1, Cebpd, Lingo2	22	CCAAUAUUGGCUGUGCUGCUCC
mmu-miR-1949	2.231649	3.982245	Ubr3,Hif1a, Trim2, Taok1, Tgfbrap1, Rb1, Tgfb2	24	CUAUACCAGGAUGUCAGCAUAGUU
mmu-miR-6952-3p	3.362607	3.965036	Cemip, Rab1, Nova1, Lingo2, Nomo1, Ikbkb, Mtus1, MMP20, Sp1, Cdk1	22	UCUCUGACUCUGCCUCCCACAG
mmu-miR-433-5p	1.776922	3.873063	Smad3, Pak3, Nav2, Smad9, Il1r1, Cdk12	22	UACGGUGAGCCUGUCAUUAUUC
mmu-mir-3069	1.598601	3.821933	Crem, Lamp5, Api5, Mbp(77%), Neta1	65	CUUGGCAGUCAAGAUAUUGUUUAGCAGACGGAGCGGUUUCUGUUGGACACUAAGUACUGCCACAA
mmu-miR-350-3p	2.117072	3.709891	Epcam, Pak6, CD163, Ilf3, Ctbp1, Mdh, E2f5, Gbas, Cxxc5, Mapk9, Nedd9	22	UUCACAAAGCCCAUACACUUUC
mmu-miR-539-5p	1.709518	3.523559	Mpz(56%), Smad5(52%), Enc1, Gas2, Map3k2, Cd44, Sp1, Dab2, Egfr	22	GGAGAAAUUAUCCUUGGUGUGU
mmu-miR-155-5p	6.260639	3.361842	Nova1, Cebpb, Smad2(85%), Tab2, Hif1a, Kras, Il6ra, E2f2, Il7r	23	UUAAUGCUAAUUGUGAUAGGGGU
mmu-miR-7050-5p	1.581792	3.325942	Cxcl16, Cdc14b, Ikbkb, Vdr, Nrg3, Map3k9, Nav1, Usp7, Scai, Cxcl14, Tgfb2, Smad2(52%)	21	ACAGGAGAAGGGGGUGAGAGA
mmu-miR-412-3p	2.145554	3.249526	Rims2, Clock, Mapk9, Taok1, Cdc42, Hip1, Tgfb3	20	UUCACCUGGUCCACUAGCCG
mmu-miR-6910-5p	1.835852	3.243152	Irs1, Lrrc36, Neurog1, Mpzl2, Cnnm1, CD40lg	23	UGGGGGUAGGGCACCAGUGGGCA
mmu-miR-6971-5p	2.145539	3.175284	Myrf, Map4, Vamp2, Taok3, Gas7, Notch4, Sp1, Nav2, Cadm4, Nova2, Neurog2, Lingo1, Wnt3, Sox9, Cadm3, Sp2, Ndrg2, Casp9, Il25, Cstb, Cd69, Notch1, Runx1, Cd7, Cx3cl1, Neurog1, Notch3	20	UGGGGGAGGGUGUAGAGGCU
mmu-miR-7674-5p	2.17967	3.059262	Akt3, Cdh2, Cadm1, Il6ra, Cd37, Mpzl1	24	UGAGGUGUGGGCAGCAUGAGGACU
mmu-miR-669a-5p	2.361674	3.041107	Cdk17, Has2, Bcor, Cdk13, Cd86, Bcl2, Cend1, Rad52, Tgfbrap1	24	AGUUGUGUGUGCAUGUUCAUGUCU
mmu-miR-467d-3p	3.182543	2.803507	Rtn4(63%), Fgfr2, Tgfbr3, Bmper, Cxcl5, Ccnc, Tab2, Cxcl16, Smad9, Mpzl1	22	AUAUACAUACACACACCUACAC
mmu-miR-199a-3p	2.686187	2.772667	Cd151, Nova1, Pak4, Fn1, Map3k4, Cdk17, Mal2, Sp1, Tack1, Rb1, Id4, Sumo3, Cxxc5, Mtor, Net1, Nedd4, Casp9, Vamp3, Bcl3	22	ACAGUAGUCUGCACAUUGGUUA
mmu-miR-199b-3p	2.686187	2.772667	Cd151, Itga3, Serpine2, Sp1, Rb1, Sumo3, Cxxc5, Map3k5, Mtor, Ndrg1, Casp9, Tab3, Ccl7, Ceacam12	22	ACAGUAGUCUGCACAUUGGUUA
mmu-miR-466a-3p	1.584525	1.937278	Omg(98%), Rtn4(84%), Smad7(80%), Smad1(78%), Mpzl1(78%), Cxcl1, Cxcl12, Cxcl16	23	UAUACAUACACGCACACAUAAGA
mmu-miR-466e-3p	1.584525	1.937278	Omg(98%), Rtn4(84%), Smad1(78%), Mpzl1, Wnt3, Cxcl1, Esm1, Cxcl2, Ndrg4, Cxcl16	23	UAUACAUACACGCACACAUAAGA
mmu-miR-467f	2.979231	1.720809	Gabrb3, Nova1, Ube2b, Mmp11, Mcam, Cxcl16, Smad7, Pten, Smad6, Zeb1, Smad9, Id2, Mcl1	21	AUAUACACACACACACCUACA
mmu-miR-3087-5p	1.624662	1.426936	Hap1, Klk11, Gas7, Stim1, Lingo1(92%), Ncam2, Cxcl16, Cav1, Smad3, Smad10, Ceacam1, Runx3	21	CAGGGCAGGGCAAGAGUUGAG
mmu-mir-466b-5	1.581792	0.824707	Sp1, Bcor, Hdac9, B2m(62%), Cxcl12(57%), Cd53, Akt3, Neurod1, Api5	88	UGUGUAUGUGUUGAUGUGUGUGUACAUGUACAUGUGUGAAUAUGAUAUACAUAUACAUACACGCACACAUAAGACACAUAUGAGCACA

**Table 2 ijms-21-00646-t002:** Downregulated miRNA in a cuprizone-induced demyelination mouse model (Fold Change ≥ 1.5).

Transcript ID (Array Design)	CPZ_5/Control.fc	CPZ_5/Control.volume	Predicted Targets in miRDB	Sequence Length	Sequence
mmu-miR-145a-5p	−1.55682	10.8367	Mpzl2, Rin2, Myrf, Smad3(85%), Twist2, Socs7, Mpzl1, Tgfa	23	GUCCAGUUUUCCCAGGAAUCCCU
mmu-miR-219a-2-3p	−2.83282	8.879056	Mthfd2l, Mapk8, Sdc2, Plk2, Cd36	22	AGAAUUGUGGCUGGACAUCUGU
mmu-miR-338-5p	−1.5951	7.399533	wif1, Cav2, Nedd1, Chl1, Sox6, Sp2, Snai1, Fasl, nanog, Tgfbr1	22	AACAAUAUCCUGGUGCUGAGUG
mmu-miR-200c-3p	−49.2278	6.603223	Zeb1, Zeb2, Nova1, Cdk17, Tbk1, jun, Mdm4, Vegfa, Ets1, Hif1a, Il17a, Mmp12, Casp2, Sdc2	23	UAAUACUGCCGGGUAAUGAUGGA
mmu-miR-182-5p	−53.6342	6.358183	Mtss1, Smad1, Bnip3, Vamp3, L1cam, Bmper, Mmp8, Smad4, Olig3, Rtn4(62%), Bdnf	25	UUUGGCAAUGGUAGAACUCACACCG
mmu-miR-700-5p	−1.5113	5.970658	Nav2, Rb1, Notch1, Plk3, Scai, Mapk1, Smad1, Hdac7, Smad2	22	UAAGGCUCCUUCCUGUGCUUGC
mmu-miR-7018-5p	−1.83096	5.038515	Lingo2, Jak2, Cpm(carboxypeptidase M), Dnm3, Rb1, Il10ra	24	GUGAGCAGACAGGGAGUGGUGGGG
mmu-miR-206-3p	−1.95263	4.988861	Ets1, Vamp2, Caap1, Hsp90b1, Cav2, Clock, Bdnf, E2f5, Sp2, Wnt3, Ndrg1, Mpl, Kras, Snai2, Nedd9	22	UGGAAUGUAAGGAAGUGUGUGG
mmu-miR-484	−1.70343	4.967894	Hnf1a, Lamb3, Il21r, Ncan, Clock, Klkb1, Gdnf	22	UCAGGCUCAGUCCCCUCCCGAU
mmu-miR-669n	−1.64951	4.877126	Mog(61%), Mobp(53%), Ncam2, Braf, Mapk8, Il8r1, Bmp4, Clock, Cxcr6, Tgfb2, Src, Cadm2	20	AUUUGUGUGUGGAUGUGUGU
mmu-miR-193a-5p	−2.45265	4.84976	Smad9, Nova1, Trim2, Usp4, Adam22	22	UGGGUCUUUGCGGGCAAGAUGA
mmu-miR-466i-5p	−1.68084	4.711763	Nrxn1, Lrrc32, Ceacam2, Neurod2, Smad7, Cxcl15, Bmp4, Zeb2, snai2, Olig2, Smad2(59%), Atm, Timp2, Nav1, Tgfbr2, Cpm	20	UGUGUGUGUGUGUGUGUGUG
mmu-miR-6937-5p	−1.77373	4.572184	Bmi1, Nrg2, Mcl1, Zeb2, Mtss1, Bri3	24	UAGCUGUAAGGGCUGGGUCUGUGU
mmu-miR-7224-3p	−1.6003	4.110822	scamp1, Taok1, Rock1, Tgfb2, Mob3a, Adam2, Pak7, Gas7, Sumo1, Nova1, Mdm2	21	UCCACUGAGAGGACCACCCAC
mmu-miR-1906	−1.54236	4.108277	Vav2, Map3k4, Cdk17, Bai3, Smad3(96%), Lingo1(60%), Il18bp, S100b, Mmp2	22	UGCAGCAGCCUGAGGCAGGGCU
mmu-miR-669k-5p	−2.07811	4.028569	Serpina3n(64%), Rtn4(82%), Scai, Bmp7, Bean1, Clock, Kras, Irf1, Cpd, Mapk10, Mmp16, Neurod6, Akt3	25	UGUGCAUGUGUGUAUAGUUGUGUGC
mmu-miR-3082-5p	−1.55267	4.023537	Lingo2(60%), Alx1, Nova1, Anxa1, Gsk3b, Nrcam, Bcor, Ccng2, Gas7, Bai2, Ceacam2, Il18r1	22	GACAGAGUGUGUGUGUCUGUGU
mmu-miR-3069-3p	−1.69083	4.006251	Mbp(77%), Rtn4(57%), Irf2, Hiat1, Lamp5, Cadm1, Cd93, Map3k3, Pdpn	22	UUGGACACUAAGUACUGCCACA
mmu-miR-7016-5p	−2.04745	3.821594	Nova2, Gpr173, Taok3, Nes, Bak1, Klk4, Ncan, Cd44, Kit, S100a16, Cxcl10, Vegfa, Ngfr, Nrp	21	CAGGGAGGGGAGCGAGAGUAG
mmu-miR-466f-3p	−1.81594	3.785958	Smad6(66%), Smad7(59%), Lingo2(58%), Smad2(56%), Smad9(53%), Cd81, Id2, Scai, Mmp11, Adam2, Mmp12	21	CAUACACACACACAUACACAC
mmu-miR-93-3p	−1.7332	3.729338	Sox6, Smad2(95%), Smurf2, Hif1a, Smad5(66%), Sumo1	22	ACUGCUGAGCUAGCACUUCCCG
mmu-miR-200b-3p	−77.6433	3.668315	Zeb2, Zeb1, Nova1, Cdk17, Mmd, Jun, Pak7, Ets1, Rock2, Nedd1, Hif1a, Wnt1, Mmp12, Egfr	22	UAAUACUGCCUGGUAAUGAUGA
mmu-miR-200a-3p	−54.4375	3.527988	Rtn4rl1(58%), Tgfb2, Ccne2, Zeb1, Snip1, Socs7, Mapk8, Serpinh1, Clock, Egfr, Cadm1	22	UAACACUGUCUGGUAACGAUGU
mmu-miR-3097-5p	−1.56809	3.520421	Smad2(61%), Olig2, Lamp5, Fap, Nlk, Taok1, Mapk10, Tgfbrap1, Neurog2	23	CACAGGUGGGAAGUGUGUGUCCA
mmu-miR-8100	−1.68734	3.498058	Smad7(95%), Lrrc59, Vamp2, Scamp2, Nova2, H1f0, Ncan, Hip1	23	AGGAGGAAAGGGAGCAAGCAGGU
mmu-mir-7017	−1.69083	3.42259	Smad3(50%), Tgfbr2(90%), Erbb2, Il6st, S100a14, Kit, Cd40, Dnmt3a, Mmp24, Il11	62	GUCCCAGAGGGUUGUGAGACUAGGGCUGUGCUUCCUGCCUAACCCUGCUCCUCUCCCUCCAG
mmu-miR-429-3p	−31.3886	3.334322	Zeb2(100%), Zeb1(100%), Nova1, Cdk17, Jun, Pak7, Vegfa, Bap1, Taok1, Nedd1, Map3k9, Wnt1, Dnmt3a	22	UAAUACUGUCUGGUAAUGCCGU
mmu-miR-200b-5p	−42.3644	3.3329	Sp3, rab1, wnt5a, Il7, Trim2, Mapk8	22	CAUCUUACUGGGCAGCAUUGGA
mmu-miR-32-3p	−3.57328	3.036032	Socs6, Jam3, Nek9, Kras, Mmd, Lyn, Lrrc39, Ndnf, Nova1, Zeb2(72%), Cd84, Bnip3l	21	CAAUUUAGUGUGUGUGAUAUU
mmu-miR-3473f	−4.44237	2.468359	Pmp22, Il1b, Kat6b, Socs1, Gfap(76%), Zeb2(68%), Fgfr1, Neurod1	20	CAAAUAGGACUGGAGAGAUG

## Data Availability

The miRNA array data obtained in this study have been deposited in the NCBI’s Gene Expression Omnibus (GEO) repository and are accessible through the GEO Series accession number GSE122839 (https://www.ncbi.nlm.nih.gov/geo/query/acc.cgi?acc=GSE122839). Other datasets used and/or analyzed during the current study are available from the corresponding author on reasonable request.

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
