# Peer review of "Differential Expression of miRNAs and Behavioral Change in the Cuprizone-Induced Demyelination Mouse Model"

_ijms, 2020, doi:10.3390/ijms21020646_

Round 1
Reviewer 1 Report
The work of Seung Ro Han et al. investigates behavioral and miRNA changes in the standard model of cuprizone-induced demyelination\remyelination. The approach based on miRNA analysis in this standard model is relatively new and interesting. The manuscript also includes a set of behavioral tests that are performed relatively rarely in this model. Despite the large amount of the obtained results, the manuscript is not solid enough and has no single specific goal. Moreover, it contains flaws in the description of the methods, presentation and interpretation of the results.
Two large parts of the results concerning behavioral data and miRNA analysis seem rather like separate blocks poorly connected to each other. Besides, behavior was explored at different time points including remyelination, whereas miRNA analysis was not performed at the endpoint. This inconsistency conflicts with the title of the article and confuses the reader. In the Method section, the general design of behavioral testing including time points within the experiment, as well as the sample size of the groups and time points, is missing. From fig.1 we see two time points of collection of brain samples. However, it is unclear how many brains out of 6 were collected and how many mice were tested behaviorally at the endpoint. Even more confusing is the legend for the figure, which states that body weight for each group was measured for 5 animals throughout the experiment. Also, the authors specify only two time points of collection of brain tissue and it is unclear how the authors performed RT-PCR and WB without the remyelination group. Statistical analysis: ANOVA with Greenhouse-Guesser correction and following post-hoc analysis (Bonferroni, Tukey, or LSD tests) are more suitable for the experiment with several groups and time points. They take into account the design with multiple measurements unlike multiple t-tests. Quantification of optical density of LFB and MBP staining is unclear. It is poorly described in the Method section. Besides, in fig.5 we see that the both stains include nuclei. Usually, studies exclude nuclei from analysis by not staining them (Khodanovich et al., 2017, Sc rep) or extracting them from photos (Underhill et al., 2012, Neuroimage). Otherwise, what optical density is quantified, myelin or myelin+nuclei? The photos in Fig. 5, especially LFB staining, seem out of focus; separate fibers are difficult to recognize. The interpretation of the results for miRNAs is very selective in favor of the authors’ own hypothesis. For example, with respect to the NgR gene. Its expression is increased, and the authors relate this to an increase in miR 20a-5p and miR 155-5p miRNAs, which suppress translation of Smad2, Smad4, and TGFβR1 (TGFβR1 in the text, although TGFβR2 in the text is a misprint?). An alternative name for the NgR gene is Rtn4rl1. Tables 1 and 2 have two micro-RNAs that suppress the Rtn4rl1 (NgR) gene - mmu-miR-34c-5p and mmu-miR-200a-3p. Mmu-miR-34c-5p increases in the demyelination group (should therefore reduce Rtn4rl1 (NgR)). Mmu-miR-200a-3p decreases in demyelination (Rtn4rl1 (NgR) should increase). Therefore, the observed effect of increasing Rtn4rl1 (NgR) can be associated with a decrease in suppression from mmu-miR-200a-3p rather than just the indirect effect of increasing miR 20a-5p. A similar situation is with the Id1 gene. Its expression is reduced in demyelination. The authors again speculate that the effect is associated with the action of miR 20a-5p and miR 155-5p on Smad and NgR. However, in table 1 there is mmu-miR-195a-3, which suppresses Id1 and it increases with demyelination. Why not assume that mmu-miR-195a-3 caused the effect on Id1? Perhaps the discussion should not ignore such direct connections. Proteins Smad2, Smad3, Smad4, as seen from the tables, are inhibited by various miRNAs. Probably, the authors should discuss these relationships, and not adhere to only one hypothesis.
Reviewer 2 Report
This paper details the changes in miRNA expression following treatment in CPZ in relation to demyelination and remyelination. Behavioural assessment and histology were also performed. The paper details 240 miRNA changes and some were investigated further in relation to demyelination. However, the paper is poorly designed and numerous problems arise as a result.
Introduction:
The intro talks about MS. Cuprizone is NOT a model of MS as it compromises the peripheral immune system (see Sen M et al 2019). It is useful to study demyelination and remyelination processes. Therefore, the opening statement of the introduction should be modified. Moreover, there is little evidence that CPZ is a neurotoxicant or induces neuronal cell death, so this statement needs modifying and relevant references cited. The mechanisms of CPZ action are known to be associated with copper chelation and mitochondrial dysfunction and a recent proteomics analysis supports this (Sen et al. 2019).
The introduction needs a testable hypothesis; it is not clear what the purpose of this work is. A hypothesis is presented on p10 in the results section but this should be stated in the introduction.
Methods: This whole section needs revamping.
How many animals were used in the study as a whole and in the different individual experiments? It appears that two runs of CPZ were used with n’s of 6 & 7 but these are insufficient numbers for the different aspects of the work. Where is the power calculation? MS preferentially affects females yet only males were used, why? Was the cuprizone made fresh each day? Why was no baseline behaviour performed for the rotarod apparatus (see fig 2)? Why did it only start at 6 weeks and why on two consecutive days? Falls & flips were recorded but only falls were described in the results Fig 2. Why was grip strength measured? What evidence of motor neurotoxicity is presented to substantiate the claim? Again, where is the baseline data & how many animals were used? Same comments for open field assessment. Baseline, number of animals, frequency of testing.Why use the corpus callosum as a proxy for demyelination in motor systems? The corpus callosum is a commissural pathway and not a projection pathway from the motor cortex. Histology of the relevant motor areas should be demonstrated so as some anatomical correlation can be made with the behaviour. Please provide a reference for the LFB method. What concentrations of blocking solution and horse anti mouse IgG? Additionally, how long were the sections developed in DAB? Time differences can lead to intensity differences and since intensity was used as a measure of demyelination, more details are required. Likewise further detail of how the intensity of LFB/MBP are needed e.g. number of sections per animal, number of animals etc. P15. Please explain how the fold changes were calculated and explain why and what the Zeuss program was used to analyse. The description of the western blot method is inadequate and further details rea needed.
Results
Because of the small n values, none of the statistics are believable in any of the figures. What is obvious from Fig 1 is that CPZ don’t lose weight but don’t increase weight until after cessation of CPZ. Please explain why the experiment was repeated in SFigs 1-3 and why this data is not part of the main manuscript. This suggests poor experimental design. The details of rotarod behaviour are inadequately described. No baseline data is described and there are no details of the flip behaviour data as described in the methods. The data from the open field graphs show no significant differences in any parameter measured. The results need to be rewritten to reflect this. Statement on lines 154-5 is incorrect. There is no evidence that demyelination contributes to the purported behavioural changes. Lines 167-70 are not results they are part of the discussion Lines 215-219 is not a result but contains a hypothesis so is more relevant to the introduction.Figures
Figure 1: The number of animals needs to be corrected. A says 6 B says 5. Moreover, 2 animals/group were sacrificed at different times meaning behavioural tests are based n”s of 4, 2 & 2. This is insufficient to do statistics.
Why was not a 2 way repeated measures ANOVA not used in B? This is the correct test.
Figure 2: why are the individual trials shown? Surely, the correct analysis is the average from the 3 trials? Moreover, the N’s are 2 in B-D, F-H, not 5 as stated in the legend.
Figures 3 & 4, 7 & 8: Same issues with N values in this figure
Fig 5; n values. SD bars in graphs missing
Figure 6 explain colour coding scale in Figs A & C and state it is a log2 scale in the legend.
The discussion is inadequate and needs a rewrite to more accurately reflect the limitations of the results.
